

# Time of night and moonlight structure vertical space use by insectivorous bats in a Neotropical rainforest: an acoustic monitoring study

Dylan G.E. Gomes[1], Giulliana Appel[2] and Jesse R. Barber[1]

[1] Department of Biological Sciences, Boise State University, Boise, ID, USA
[2] Programa de Pós-graduação em Ecologia, Instituto Nacional de Pesquisas da Amazônia, Manaus, Brazil

## ABSTRACT

**Background:** Previous research has shown diverse vertical space use by various taxa, highlighting the importance of forest vertical structure. Yet, we know little about vertical space use of tropical forests, and we often fail to explore how this three-dimensional space use changes over time.

**Methods:** Here we use canopy tower systems in French Guiana and passive acoustic monitoring to measure Neotropical bat activity above and below the forest canopy throughout nine nights. We use a Bayesian generalized linear mixed effect model and kernel density estimates to demonstrate patterns in space-use over time.

**Results:** We found that different bats use both canopy and understory space differently and that these patterns change throughout the night. Overall, bats were more active above the canopy (including *Cormura brevirostris*, *Molossus molossus*, *Peropteryx kappleri* and *Peropteryx macrotis*), but multiple species or acoustic complexes (when species identification was impossible) were more active in the understory (such as *Centronycteris maximiliani*, *Myotis riparius*, *Pteronotus alitonus* and *Pteronotus rubiginosus*). We also found that most bats showed temporally-changing preferences in hourly activity. Some species were less active (e.g., *P. kappleri* and *P. macrotis*), whereas others were more active (*Pteronotus gymnonotus*, *C. brevirostris*, and *M. molossus*) on nights with higher moon illuminance.

**Discussion:** Here we show that Neotropical bats use habitat above the forest canopy and within the forest understory differently throughout the night. While bats generally were more active above the forest canopy, we show that individual groups of bats use space differently over the course of a night, and some prefer the understory. This work highlights the need to consider diel cycles in studies of space use, as animals use different habitats during different periods of the day.

## INTRODUCTION

The study of space use has long interested ecologists (*Elton, 1927*), and more recently three-dimensional space use has been shown to be important for many taxa including

Corresponding author
Dylan G.E. Gomes,
dylangomes@u.boisestate.edu

arthropods (*Schulze, Linsenmair & Fiedler, 2001*; *Basset et al., 2003*), birds (*Pearson, 1971*; *Walther, 2002*), rodents, marsupials (*Vieira & Monteiro-Filho, 2003*) and bats (*Francis, 1994*; *Bernard, 2001*). Understanding space use over time is vital if we hope to accurately assess habitat use and quality (*Bernard, 2001*; *Müller et al., 2013*; *Appel et al., 2019*). This is especially true in the tropics where biodiversity loss from deforestation is high (*Laurance, 1999*; *Giam, 2017*).

Bats are ideal study organisms for exploring vertical stratification of space-use. The ability for powered flight allows them to easily access the various strata of the forest, and previous studies have shown that Neotropical species vary in their use of three-dimensional space (*Kalko & Handley, 2001*; *Pereira, Marques & Palmeirim, 2010*; *Rex et al., 2011*; *Marques, Ramos Pereira & Palmeirim, 2016*). However, most of these studies have used mist-nets, which are more likely to capture bats in the Family Phyllostomidae (*Kalko & Handley, 2001*; *Pereira, Marques & Palmeirim, 2010*; *Rex et al., 2011*). Given that most bats in Neotropical rainforests are not phyllostomid bats, but rather aerial insectivores from other Families, there is a gap in vertical stratification knowledge within these forests (*Marques, Ramos Pereira & Palmeirim, 2016*; *Silva et al., 2020*). Aerial insectivorous bats rely on echolocation to orient, navigate, and forage on the wing for arthropod prey (*Schnitzler, Moss & Denzinger, 2003*). Echolocation calls of aerial insectivorous species are generally distinct to the species level, which allows bats to be relatively easily monitored. Passive acoustic monitors are rapidly becoming low-cost and open-source, and advances in automatic detection of biotic signals (i.e., echolocation calls) have greatly increased analytical throughput (*Gibb et al., 2019*; *López-Baucells et al., 2019*).

Passive monitoring of rainforest bats during the dry season in Brazil suggests that bat activity and species diversity is higher in the canopy, relative to mid- or below-canopy (*Marques, Ramos Pereira & Palmeirim, 2016*). In *Marques, Ramos Pereira & Palmeirim (2016)*, only one species (*Myotis riparius*) did not prefer to forage above the canopy. This may be due to the species' aversion to moonlight (*Appel et al., 2019*), which would likely be exacerbated above the canopy. The result that all other bats prefer to forage above the canopy may be a result of high insect abundance in the canopy (*Basset et al., 2003*). Many nectar feeding Lepidoptera (e.g., Sphingidae), for example, are more abundant high in the canopy, where more flowers are present (*Schulze, Linsenmair & Fiedler, 2001*). Yet, it is likely that the abundance of arthropod prey, and thus bats foraging above the canopy, would vary throughout the night. Indeed, some tropical insectivorous bat species adjust their activity during the night to take advantage of more favorable periods to forage, such as to avoid rain or moonlight (*Appel et al., 2019*).

Little is known about temporal patterns of vertical space use of aerial insectivorous bats, and surprisingly little is known about bats in the Guiana Shield. Here we use passive acoustic monitors to survey vertical space use by Neotropical bats in French Guiana to fill this knowledge gap. Since only one of the bat species (*Myotis riparius*) we detected in this study has previously been found to prefer forest understory (*Marques, Ramos Pereira & Palmeirim, 2016*), we expected to see similar results to previous work, where

most bats prefer the forest canopy. Yet, if we are to make generalizable inference, it is important to validate past results in different areas across different times of year, such as we attempt to do here. Additionally, we aim to explore how bats use this vertical space over the course of a night, which may highlight that different strata of the rainforest are important during different times.

# METHODS

## Setup

We worked at the Saut Pararé Nourages research station (4°2′30″ N, 52°40′30″ W), French Guiana from the evening of 10 April 2018, to the morning of 19 April 2018 in the wet season. The area contains a dense, nearly undisturbed old-growth rainforest, dominated by Burseraceae trees. This location has a humid climate (300 cm of precipitation per year) with a short dry season from mid-August to mid-November with less than 10 cm of rainfall (*Joetzjer et al., 2017*). Mean monthly air temperature in this location ranges from 25.5 °C in January to 27.5 °C in October (*Obregon et al., 2011*). We sampled above and below the forest canopy at two canopy towers (180 m apart), which are a part of the COPAS infrastructure (*Gottsberger & Döring, 1995*; *Gottsberger, 2017*). Canopy towers were 45 m high (*Gottsberger, 2017*), placing them above the nearby forest canopy which was approximately 40 m high or less (*Joetzjer et al., 2017*).

We conducted paired sampling on top of and below the canopy towers, to get a measure of bat activity above the forest canopy and within the forest understory, respectively. At each sample site, we deployed a passive acoustic monitoring unit (Song Meter SM3) with an omnidirectional ultrasonic microphone (SMU; Wildlife Acoustics, MA, USA). To reduce echoes in recordings, microphones on canopy towers were attached to the ends of 2 m poles, which in turn were attached to the COPAS platform. Thus the above-canopy microphones were 45 m off the ground. The passive acoustic monitoring units in the forest understory were placed in relatively open, non-dense flyways that had little vegetation. The microphones were also attached to the ends 2 m poles as above, which were in turn attached to the trunks of trees, 1.5 m off the ground, far from any vegetation to reduce echoes. The absolute detection range of the microphones is dependent on many factors including sonar emission intensity, ambient background sound levels and the frequency of the signal. Conservative detection estimates are between 8 and 30 m for a sonar emission of 100–120 dB, a peak frequency of 30–50 kHz, and a 20 dB background sound level, with no interfering vegetation (*Agranat, 2014*). Further, the vegetation of the forest canopy formed a barrier to the transmission of the short ultrasonic wavelengths of bat echolocation between the recorders. Thus it was exceedingly unlikely that both acoustic monitors simultaneously detected bats both above and below the canopy. We left passive acoustic monitors in the field for the duration of the study, and they were programmed to automatically turn on at sunset and off at sunrise (12 h per night × 9 nights × 4 locations = 432 monitored hours) and to record with a 16-bit depth, 384 kHz sample rate, with an internal 16 kHz high pass filter, and a 1.5 ms minimum trigger duration.

**Table 1 Passive acoustic monitoring observations over nine nights within the understory and canopy at the COPAS facility in French Guiana.**

| Acoustic group | Understory | Canopy | Total |
|---|---|---|---|
| *Peropteryx trinitatis* | 0 | 1 | 1 |
| *Pteronotus sp.* | 0 | 1 | 1 |
| *Saccopteryx gymnura* | 1 | 0 | 1 |
| *Diclidurus sp.* | 2 | 3 | 5 |
| *Molossus molossus* | 0 | 20 | 20 |
| *Pteronotus gymnonotus* | 2 | 19 | 21 |
| *Pteronotus rubiginosus* | 20 | 15 | 35 |
| *Lasiurus blossevilli/Rhogeessa io* | 0 | 37 | 37 |
| *Lasiurus* sp. | 69 | 3 | 72 |
| *Phyllostomidae* | 13 | 84 | 97 |
| *Myotis riparius* | 203 | 2 | 205 |
| *Myotis simus/nigricans* | 143 | 88 | 231 |
| *Molossidae group B* | 55 | 198 | 253 |
| *Molossidae group A* | 57 | 214 | 271 |
| *Pteronotus alitonus* | 362 | 4 | 366 |
| *Cormura brevirostris* | 10 | 379 | 389 |
| *Saccopteryx leptura* | 397 | 671 | 1,068 |
| *Peropteryx kappleri* | 280 | 1,264 | 1,544 |
| *Centronycteris maximiliani* | 1,270 | 944 | 2,214 |
| *Saccopteryx bilineata* | 1,018 | 3,512 | 4,530 |
| *Peropteryx macrotis* | 70 | 4,692 | 4,762 |

**Note:**

*Diclidurus sp.* may include *Diclidurus albus, D. scutatus*, and/or *D. ingens. Lasiurus sp.* may include *Lasiurus ega, L. castaneus, L. egregius*, and/or *L. atratus. Molossidae group A* may include *Molossus sinaloe, M. rufus, M. currentium, Promops centralis, Cynomops planirostris* and/or *C. paranus. Molossidae group B* may include *Cynomops greenhalii, C. abrasus, Eumops auripendulus, E. glaucinus, E. dabbenei, E. hansae, E. maurus, Nyctinomops laticaudatus* and/or *Tadarida brasiliensis.*

## Sonar sequence identification

Bat recordings were batch processed with Sonobatch automatic scrubbing software to exclude files that did not contain bat calls (*Szewczak, 2015*). We then visualized the remaining 16,123 sequences with Kaleidoscope Software (version 4.3.2; Wildlife Acoustics, MA, USA) and identified the calls following the libraries of Amazonian bat echolocation (*López-Baucells, 2018*) and echolocation characteristics from the literature (*Barataud et al., 2013*; *Arias-Aguilar et al., 2018*). When possible, we identified bat calls to the species level or identified the call as an acoustic complex when species-level identification was impossible (*López-Baucells, 2018*; *Torrent et al., 2018*). Our data included a total of 13 species and eight acoustic complexes, with a total of 21 sonotypes from the families Emballonuridae, Molossidae, Mormoopidae, Phyllostomidae and Vespertilionidae (Table 1). We defined bat activity as the number of bat passes per hour each night. A bat pass is a sequence of 5-s recording that has a minimum of two recognizable search-phase calls per species (*Torrent et al., 2018*; *Appel et al., 2019*).

## Statistical analysis

Data were explored following the protocol of *Zuur, Ieno & Elphick (2010)*. We built a generalized linear (mixed) effects model within a Bayesian framework with MCMC in Stan within the R programming language (*R Core Team, 2017*) package "rstanarm" and function "stan_glmer.nb" (*Gabry & Goodrich, 2016*). MCMC is essentially a simulation technique to obtain the distribution of each parameter in a model (*Zuur & Ieno, 2016*). All model settings were "rstanarm" defaults (see supplementary code or *Gabry & Goodrich, 2016*). For example, priors were weakly informed normal distributions ($\mu = 0$, $\sigma = 2.5$), the number of chains = 4 and the number of iterations = 2,000, with 1,000 warmup iterations. We visually checked model residuals (*Zuur & Ieno, 2016*) and trace plots, and all chains mixed well (see Supplement). We inspected predictors for collinearity by using Variance Inflation Factors (VIF) with the function "check_collinearity" from the package "performace" (*Lüdecke et al., 2019*) and all VIF < 2 (see Supplement). There were no divergent transitions in simulated parameter trajectories, suggesting the posterior was well-explored, nor issues with convergence (all rhat values were very close to 1; see Table 2 and Supplement). We did not thin chains (*Link & Eaton, 2012*).

We analyzed the response data, which were counts of bat passes, with a negative binomial distribution and log link function. In this model (see Table 2), we removed all bat species (or acoustic complexes) that contained five or fewer observations, since these data are not robust enough for inference, but included all other bat species/complexes. Thus, we set a random intercept for all included bat species, with random slope for hour after sunset (0-12), vertical strata (canopy vs. understory), for the interaction between the two, and for moon illuminance—that is, each of these four terms was allowed to vary by bat species. These four terms (hour after sunset, vertical strata, the interaction between the two, and moon illuminance) were also fit as fixed effects to make inferences on "all bats" overall. We included site as a random intercept, to avoid pseudoreplication (*Zuur & Ieno, 2016*), although we did not have at least five levels (*Harrison et al., 2018*). Moon illuminance was centered by the mean and scaled by two standard deviations to both improve the computability of the model and to make this directly comparable to categorical (e.g., above vs. below the canopy) predictors (*Gelman, 2008*). We included horizontal moon illumination (*Kyba, Conrad & Shatwell, 2020*) as a fixed effect to control for any influences that moon light might have on vertical bat activity (*Hecker & Brigham, 1999*; *Appel et al., 2017*), as well as any latent processes occurring over the course of the nine day study (either due to moonlight or day of the year). Moon illuminance was calculated using custom windows command line code, *sunmoon* program (Jeff Conrad *unpublished software*). The methods are similar to those of *Janiczek & DeYoung (1987)*. Sun and Moon positions are determined using the more accurate formulas of *Van Flandern & Pulkkinen (1979)*.

To further elucidate patterns of bat activity over the course of the night, we separately analyzed the 11 most common bat species or acoustic complexes (See Table 1 for list) with kernel density estimators of bat activity by hour after sunset, by vertical strata (canopy vs. understory). We did not build kernel density estimates for other species, as the
**Table 2 Output from Bayesian generalized linear mixed-effect model (negative binomial family; log-link function).**

| Variable | Estimate | SE | 80% CI | 90% CI | N Eff | Rhat |
|---|---|---|---|---|---|---|
| Intercept | −1.63 | 0.56 | [−2.34 to −0.88] | [−2.55 to −0.68] | 1,123 | 1.003 |
| Moon Illuminance | 0.08 | 0.08 | [−0.03 – 0.19] | [−0.07 – 0.22] | 1,420 | 1.002 |
| Hour | 0.02 | 0.05 | [−0.06 – 0.09] | [−0.08 – 0.11] | 897 | 1.005 |
| Above | 2.06 | 0.51 | [1.38 – 2.70] | [1.18 – 2.90] | 1,152 | 1.006 |
| Hour:Above | −0.28 | 0.06 | [−0.35 to −0.21] | [−0.38 to −0.18] | 2,233 | 1.001 |
| b[Int. \|cenmax] | 3.39 | 0.62 | [2.61 – 4.19] | [2.40 – 4.44] | 1,378 | 1.002 |
| b[Hour \|cenmax] | 0.00 | 0.07 | [−0.09 – 0.10] | [−0.12 – 0.13] | 1,791 | 1.002 |
| b[Above \|cenmax] | −1.09 | 0.70 | [−2.00 to −0.17] | [−2.29 – 0.05] | 1,759 | 1.003 |
| b[Moon Illuminance \|cenmax] | −0.12 | 0.14 | [−0.29 – 0.06] | [−0.35 – 0.13] | 2,559 | 1.002 |
| b[Hour:Above \|cenmax] | 0.06 | 0.09 | [−0.05 – 0.19] | [−0.09 – 0.23] | 2,859 | 1.001 |
| b[Int. \|corbre] | −0.91 | 0.74 | [−1.87 – 0.02] | [−2.14 – 0.29] | 1,650 | 1.001 |
| b[Hour \|corbre] | −0.12 | 0.10 | [−0.25 – 0.00] | [−0.29 – 0.03] | 2,383 | 1.001 |
| b[Above \|corbre] | 2.15 | 0.76 | [1.23 – 3.16] | [0.98 – 3.43] | 1,949 | 1.003 |
| b[Moon Illuminance \|corbre] | 0.14 | 0.14 | [−0.03 – 0.34] | [−0.08 – 0.41] | 2,209 | 1.002 |
| b[Hour:Above \|corbre] | 0.16 | 0.10 | [0.03 – 0.30] | [0.00 – 0.33] | 2,984 | 1.000 |
| b[Int. \|molmol] | −3.25 | 1.04 | [−4.65 – −1.98] | [−5.05 to −1.63] | 2,183 | 1.001 |
| b[Hour \|molmol] | −0.13 | 0.14 | [−0.31 – 0.04] | [−0.38 – 0.08] | 2,947 | 1.000 |
| b[Above \|molmol] | 1.53 | 1.03 | [0.33 – 2.96] | [0.00 – 3.40] | 2,412 | 1.002 |
| b[Moon Illuminance \|molmol] | 0.21 | 0.17 | [0.00 – 0.45] | [−0.05 – 0.53] | 3,192 | 1.000 |
| b[Hour:Above \|molmol] | 0.12 | 0.13 | [−0.03 – 0.30] | [−0.08 – 0.36] | 2,832 | 1.000 |
| b[Int. \|Mol.A] | −0.90 | 0.64 | [−1.76 to −0.08] | [−1.99 – 0.17] | 1,334 | 1.002 |
| b[Hour \|Mol.A] | 0.13 | 0.07 | [0.04 – 0.23] | [0.02 – 0.25] | 1,221 | 1.004 |
| b[Above \|Mol.A] | 0.66 | 0.68 | [−0.24 – 1.53] | [−0.52 – 1.77] | 1,608 | 1.004 |
| b[Moon Illuminance \|Mol.A] | 0.00 | 0.12 | [−0.15 – 0.18] | [−0.19 – 0.24] | 1,576 | 1.004 |
| b[Hour:Above \|Mol.A] | 0.12 | 0.08 | [0.03 – 0.23] | [0.00 – 0.27] | 2,286 | 1.001 |
| b[Int. \|Mol.B] | −2.21 | 0.78 | [−3.26 to −1.23] | [−3.60 to −0.93] | 1,688 | 1.001 |
| b[Hour \|Mol.B] | 0.28 | 0.09 | [0.17 – 0.40] | [0.14 – 0.44] | 1,722 | 1.001 |
| b[Above \|Mol.B] | 1.72 | 0.82 | [0.73 – 2.80] | [0.45 – 3.13] | 1,884 | 1.001 |
| b[Moon Illuminance \|Mol.B] | −0.08 | 0.13 | [−0.25 – 0.09] | [−0.29 – 0.15] | 2,153 | 1.002 |
| b[Hour:Above \|Mol.B] | 0.00 | 0.09 | [−0.12 – 0.12] | [−0.16 – 0.15] | 2,841 | 1.000 |
| b[Int. \|myorip] | 2.06 | 0.58 | [1.32 – 2.81] | [1.13 – 3.03] | 1,317 | 1.002 |
| b[Hour \|myorip] | −0.08 | 0.06 | [−0.16 – 0.00] | [−0.19 – 0.03] | 1,220 | 1.003 |
| b[Above \|myorip] | −4.66 | 0.88 | [−5.88 to −3.56] | [−6.26 to −3.29] | 2,414 | 1.002 |
| b[Moon Illuminance \|myorip] | 0.08 | 0.19 | [−0.17 – 0.34] | [−0.25 – 0.42] | 3,426 | 1.002 |
| b[Hour:Above \|myorip] | −0.16 | 0.15 | [−0.36 – 0.03] | [−0.45 – 0.08] | 3,753 | 1.000 |
| b[Int. \|myo.sp] | 1.10 | 0.57 | [0.35 – 1.85] | [0.14 – 2.08] | 1,237 | 1.003 |
| b[Hour \|myo.sp] | 0.03 | 0.06 | [−0.05 – 0.12] | [−0.07 – 0.14] | 1,134 | 1.004 |
| b[Above \|myo.sp] | −1.05 | 0.65 | [−1.88 to −0.20] | [−2.14 – 0.03] | 1,590 | 1.004 |
| b[Moon Illuminance \|myo.sp] | −0.04 | 0.12 | [−0.20 – 0.13] | [−0.25 – 0.19] | 3,227 | 1.000 |
| b[Hour:Above \|myo.sp] | −0.03 | 0.08 | [−0.14 – 0.07] | [−0.17 – 0.11] | 2,964 | 1.001 |
| b[Int. \|perkap] | 1.43 | 0.74 | [0.49 – 2.39] | [0.25 – 2.67] | 1,587 | 1.001 |

| Variable | Estimate | SE | 80% CI | 90% CI | N Eff | Rhat |
|---|---|---|---|---|---|---|
| b[Hour \|perkap] | 0.03 | 0.09 | [−0.09 − 0.15] | [−0.12 − 0.19] | 1,763 | 1.003 |
| b[Above \|perkap] | 1.51 | 0.75 | [0.58 − 2.51] | [0.32 − 2.81] | 1,630 | 1.002 |
| b[Moon Illuminance \|perkap] | −0.23 | 0.16 | [−0.46 to −0.04] | [−0.52 − 0.01] | 2,413 | 1.000 |
| b[Hour:Above \|perkap] | −0.12 | 0.10 | [−0.25 − 0.01] | [−0.30 − 0.04] | 2,575 | 1.001 |
| b[Int. \|permac] | 0.64 | 0.73 | [−0.28 − 1.54] | [−0.53 − 1.79] | 1,470 | 1.002 |
| b[Hour \|permac] | −0.05 | 0.09 | [−0.17 − 0.06] | [−0.20 − 0.10] | 2,101 | 1.001 |
| b[Above \|permac] | 3.39 | 0.77 | [2.44 − 4.42] | [2.19 − 4.73] | 1,828 | 1.003 |
| b[Moon Illuminance \|permac] | −0.21 | 0.18 | [−0.46 − 0.00] | [−0.53 − 0.07] | 2,826 | 1.000 |
| b[Hour:Above \|permac] | 0.03 | 0.10 | [−0.11 − 0.17] | [−0.15 − 0.21] | 3,553 | 1.000 |
| b[Int. \|phyllo] | −1.34 | 0.69 | [−2.26 − 0.40] | [−2.54 to −0.15] | 1,608 | 1.003 |
| b[Hour \|phyllo] | 0.00 | 0.08 | [−0.10 − 0.11] | [−0.13 − 0.15] | 1,830 | 1.001 |
| b[Above \|phyllo] | 1.09 | 0.72 | [0.14 − 2.05] | [−0.12 − 2.35] | 1,711 | 1.004 |
| b[Moon Illuminance \|phyllo] | −0.04 | 0.14 | [−0.24 − 0.13] | [−0.33 − 0.18] | 3,216 | 1.000 |
| b[Hour:Above \|phyllo] | 0.06 | 0.09 | [−0.06 − 0.18] | [−0.09 − 0.22] | 3,452 | 0.999 |
| b[Int. \|pteali] | 2.14 | 0.59 | [1.39 − 2.89] | [1.20 − 3.14] | 1,260 | 1.003 |
| b[Hour \|pteali] | −0.01 | 0.06 | [−0.10 − 0.07] | [−0.12 − 0.10] | 1,265 | 1.003 |
| b[Above \|pteali] | −4.51 | 0.84 | [−5.59 to −3.48] | [−5.87 to −3.21] | 2,370 | 1.001 |
| b[Moon Illuminance \|pteali] | 0.10 | 0.15 | [−0.09 − 0.31] | [−0.14 − 0.39] | 3,755 | 1.001 |
| b[Hour:Above \|pteali] | −0.20 | 0.15 | [−0.43 to −0.03] | [−0.51 − 0.02] | 3,441 | 1.001 |
| b[Int. \|ptegym] | −2.30 | 0.91 | [−3.47 to −1.14] | [−3.80 to −0.85] | 2,051 | 1.001 |
| b[Hour \|ptegym] | −0.13 | 0.13 | [−0.31 − 0.02] | [−0.36 − 0.06] | 3,316 | 0.999 |
| b[Above \|ptegym] | 0.82 | 0.86 | [−0.27 − 1.99] | [−0.58 − 2.35] | 2,513 | 1.002 |
| b[Moon Illuminance \|ptegym] | 0.18 | 0.16 | [−0.02 − 0.41] | [−0.07 − 0.49] | 3,286 | 1.000 |
| b[Hour:Above \|ptegym] | 0.11 | 0.12 | [−0.03 − 0.28] | [−0.07 − 0.34] | 3,350 | 0.999 |
| b[Int. \|pterub] | −0.15 | 0.71 | [−1.03 − 0.76] | [−1.27 − 1.04] | 1,586 | 1.002 |
| b[Hour \|pterub] | −0.11 | 0.09 | [−0.24 − 0.00] | [−0.27 − 0.04] | 2,181 | 1.001 |
| b[Above \|pterub] | −1.17 | 0.75 | [−2.14 to −0.20] | [−2.43 − 0.10] | 1,997 | 1.001 |
| b[Moon Illuminance \|pterub] | 0.05 | 0.17 | [−0.20 − 0.26] | [−0.29 − 0.32] | 2,431 | 1.001 |
| b[Hour:Above \|pterub] | −0.13 | 0.13 | [−0.33 − 0.01] | [−0.41 − 0.06] | 3,623 | 1.000 |
| b[Int. \|sacbil] | 3.65 | 0.59 | [2.91 − 4.42] | [2.70 − 4.63] | 1,264 | 1.003 |
| b[Hour \|sacbil] | −0.07 | 0.07 | [−0.16 − 0.01] | [−0.18 − 0.04] | 1,226 | 1.003 |
| b[Above \|sacbil] | −0.02 | 0.65 | [−0.89 − 0.87] | [−1.16 − 1.12] | 1,544 | 1.003 |
| b[Moon Illuminance \|sacbil] | −0.07 | 0.15 | [−0.24 − 0.14] | [−0.29 − 0.20] | 2,204 | 1.002 |
| b[Hour:Above \|sacbil] | 0.09 | 0.08 | [−0.01 − 0.20] | [−0.04 − 0.24] | 3,023 | 1.000 |
| b[Int. \|saclep] | 2.72 | 0.56 | [1.97 − 3.45] | [1.77 − 3.66] | 1,153 | 1.003 |
| b[Hour \|saclep] | −0.08 | 0.06 | [−0.16 − 0.00] | [−0.18 − 0.03] | 988 | 1.004 |
| b[Above \|saclep] | −0.70 | 0.63 | [−1.52 − 0.15] | [−1.72 − 0.40] | 1,466 | 1.004 |
| b[Moon Illuminance \|saclep] | 0.05 | 0.13 | [−0.11 − 0.23] | [−0.16 − 0.29] | 2,378 | 1.002 |
| b[Hour:Above \|saclep] | 0.02 | 0.07 | [−0.07 − 0.12] | [−0.10 − 0.15] | 2,698 | 1.000 |
| b[Int. \|las.sp] | −3.75 | 0.95 | [−5.04 to −2.57] | [−5.49 to −2.28] | 1,848 | 1.001 |
| b[Hour \|las.sp] | 0.44 | 0.10 | [0.33 − 0.57] | [0.30 − 0.62] | 1,679 | 1.001 |
| b[Above \|las.sp] | −0.72 | 1.03 | [−2.07 − 0.59] | [−2.48 − 0.99] | 2,085 | 1.000 |

*(Continued)*

**Table 2** (*continued*)

| Variable | Estimate | SE | 80% CI | 90% CI | N Eff | Rhat |
|---|---|---|---|---|---|---|
| b[Moon Illuminance \|las.sp] | −0.15 | 0.24 | [−0.48 – 0.15] | [−0.61 – 0.23] | 2,816 | 1.000 |
| b[Hour:Above \|las.sp] | −0.20 | 0.13 | [−0.39 to −0.04] | [−0.44 – 0.00] | 2,655 | 1.000 |
| b[Int. \|lasblo/rhoio] | −2.99 | 1.03 | [−4.44 to −1.73] | [−4.88 to −1.42] | 2,489 | 1.002 |
| b[Hour \|lasblo/rhoio] | −0.14 | 0.13 | [−0.33 – 0.03] | [−0.40 – 0.07] | 3,231 | 1.001 |
| b[Above \|lasblo/rhoio] | 2.12 | 1.03 | [0.95 – 3.54] | [0.62 – 3.99] | 2,944 | 1.002 |
| b[Moon Illuminance \|lasblo/rhoio] | 0.13 | 0.17 | [−0.10 – 0.38] | [−0.17 – 0.47] | 3,482 | 1.000 |
| b[Hour:Above \|lasblo/rhoio] | 0.11 | 0.13 | [−0.05 – 0.29] | [−0.09 – 0.35] | 3,116 | 0.999 |
| b[Int. \|Tower=green] | 0.21 | 0.19 | [−0.04 – 0.48] | [−0.15 – 0.59] | 3,696 | 1.000 |
| b[Int. \|Tower=red] | −0.22 | 0.19 | [−0.51 – 0.01] | [−0.63 – 0.11] | 3,631 | 1.000 |

Note:
SE, standard error, N Eff, number of effective samples in MCMC and Rhat (A.K.A. the Gelman-Rubin statistic) is a measure of how chains might be reaching different conclusions. Here, all values are very close to 1, which indicates good model convergence. The first five "Variables" are fixed effects, whereas all variables wrapped in "b[ ]" are random effects. "Int.", intercept, "Above", above canopy (relative to below the canopy; that is, the intercept), "Hour" is the hour since sunset, and everything to the right of the "|" are indicating that the effects vary by bat species/acoustic complexes and are keyed as follows: cenmax, *Centronycteris maximiliani*; corbre, *Corumura brevirostris*; molmol, *Molossus molossus*; Mol.A, molossidae group A (see Table 1); Mol.B, molossidae group B (see Table 1); myorip, *Myotis riparius*; myo.sp, *Myotis simus/nigricans*; perkap, *Peropteryx kappleri*; permac, *Peropteryx macrotis*; phyllo, Phyllostomidae; pteali, *Pteronotus alitonus*; ptegym *Pteronotus gymnonotus*; pterub, *Pteronotus rubiginosus*; sacbil, *Saccopteryx bilineata*; saclep, *Saccopteryx leptura*; las.sp, *Lasiurus* sp. (see Table 1); lasblo/rhoio, *Lasiurus blossevilli/Rhogeessa Io*.

number of counts for those species was low, and we did not feel comfortable making inferences on minimal data.

Throughout the results we report model estimates and 80% and 90% credible intervals (for all in-text estimates see R code). While these choices (including 95%) are always largely arbitrary, we chose these values because 80% and 90% intervals both display a wide interval spanning a high probability range of parameter values, especially with the 80% interval replacing the common Stan default of 50% (*McElreath, 2020*). We avoid using a 95% credible interval for a number of reasons. Firstly, these can often be misinterpreted as 95% confidence intervals (*McElreath, 2020*). The latter, in contrast to Bayesian credible intervals, assume that the interval is random and the parameter is fixed, rely on imaginary resampling of data, and are often interpreted as a hypothesis test (*McElreath, 2020*). Secondly, both 80% and 90% credible intervals reduce concerns with the computational stability of wider (e.g., 95%) intervals. In the following text we generally use 80% CI to suggest broad-scale trends, whereas we use 90% CI in the reporting of parameter estimates, to give a narrower estimate band, with higher certainty. As these are not hypothesis tests, these credible intervals give the reader a summary of the posterior distribution, thus reporting multiple credible intervals, rather than just one, help to demonstrate the shape of the posterior distribution (*McElreath, 2020*).

## RESULTS

There were 12,151 bat passes above the canopy and 3,972 below the canopy. After accounting for repeat sampling of species, hour after sunset, and moon illuminance, generalized linear mixed effects models suggest that bat activity was 9.5 times (90% CI [4.3–21.1]) higher above the canopy, relative to the understory (Table 2). Yet, patterns

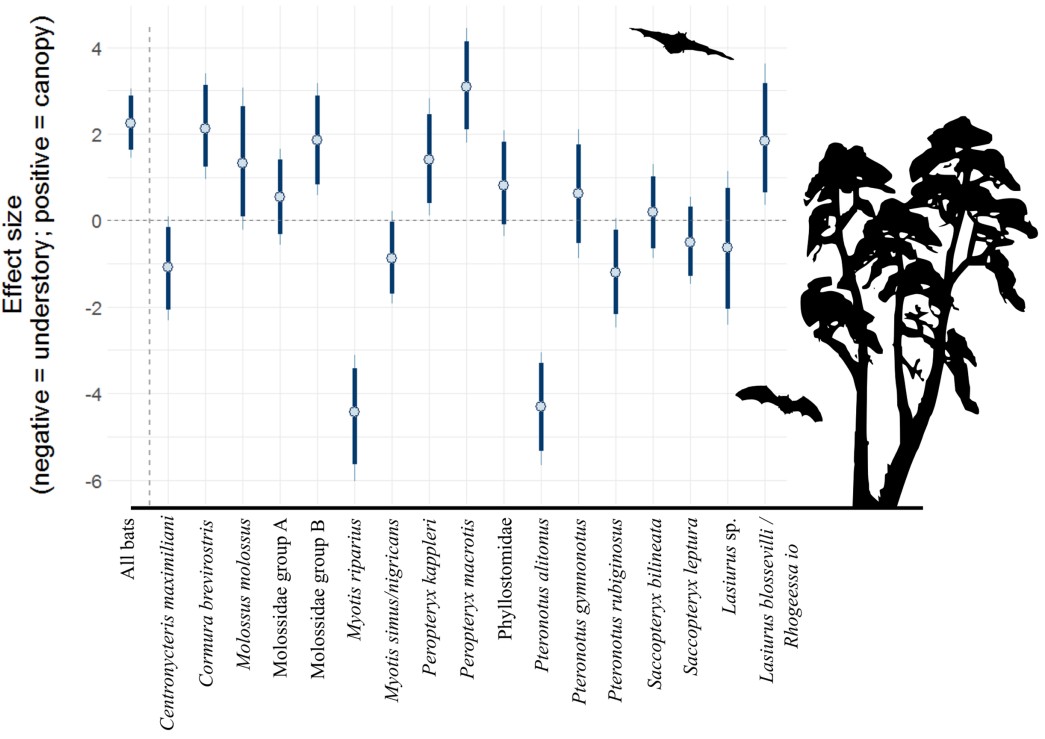

**Figure 1 Model coefficient estimates for activity in vertical strata, by bat species/complex.** Positive values on y axis indicate that bats were more active in the canopy, whereas negative values indicate that bats were more active near the forest floor. Bold lines are 80% credible intervals, whereas thin lines are 90% CI. See Table 1 caption for acoustic complex species breakdown.

for individual species (or acoustic complexes) were mixed (Fig. 1; Table 2). Broad patterns at 80% credible intervals suggest six species/complexes were more active above the canopy (*Cormura brevirostris, Molossus molossus*, Molossidae group B, *Peropteryx kappleri, Peropteryx macrotis* and *Lasiurus blossevilli/Rhogeessa Io*), five in the understory (*Centronycteris maximiliani, Myotis riparius, Myotis simus/nigricans, Pteronotus alitonus* and *Pteronotus rubiginosus*), and six show no difference (Molossidae group A, Phyllostomidae, *Pteronotus gymnonotus, Saccopteryx bilineata, Saccopteryx leptura* and *Lasiurus* sp.). Of the strongest trends, *P. macrotis* was 21.8 times more likely to be found above the canopy (90% CI [6.01–84.6]), whereas *M. riparius* was a factor of 132.8 more likely to be in the understory (90% CI [31.2–586.6]).

Overall bat activity decreased 22.0% (90% CI [14.8–29.6]) for every hour above the canopy as the night progressed, whereas activity in the understory did not change over time (90% CI [−8.2 to 10.7]). Individual bat species/complexes differed in their activity above and below the canopy as the evening progressed, depending on the species/complex (Fig. 2; Table 2). Three bat complexes increased understory use over the night, whereas none of them decreased their use of that space over time (90% CI). The *Lasiurus* sp. complex, for example, was 52.5% more active in the understory (90% CI [32.4–83.1]), each hour of the night (Fig. 2). Above-canopy use throughout the night, however, increased for two groups, and decreased for one at the 90% CI, but trended that direction

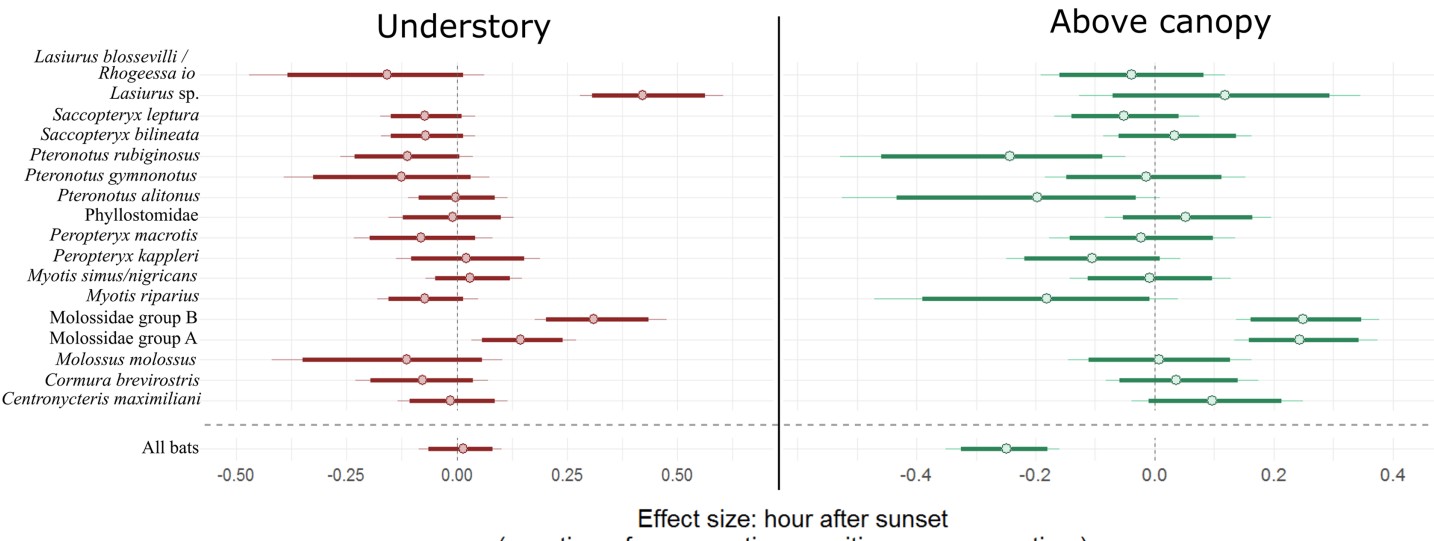

**Figure 2** **Model coefficient estimates for activity over the course of the night by bat species.** Estimates on left are for understory activity, whereas those on right are for canopy activity. Positive values on *x* axis indicate that bats were more active as time passed within a night, whereas negative values indicate that bats were more active earlier in the night. Bold lines are 80% credible intervals, whereas thin lines are 90% CI. See Table 1 caption for acoustic complex species breakdown.                                 

for two other groups (80% CI; Fig. 2). Two of the complexes (Molossidae group A and B) increased the use of both understory and above the canopy throughout the night.

*Centronycteris maximiliani* activity showed a peak of activity in the middle of the night. This species was slightly more active in the understory, relative to above the canopy, during early and late parts of the night, whereas they were more active above the canopy during the middle of the night (Fig. 3A). *S. bilineata* had higher activity in the understory at the beginning and end of the night (dusk and dawn), and higher above-canopy activity in the early-middle of the night (Fig. 3B). Both *P. kappleri* and *P. macrotis* were far more active above the canopy (relative to understory) early in the night, but there was a spike in understory activity late in the night (Figs. 3C and 3D).

There is an 80.9% probability that moonlight had a positive effect on overall bat activity. Similarly, *M. molossus*, *C. brevirostris* and *P. gymnonotus* have high probabilities of positive effects of moonlight on species activity (90.3%, 85.4% and 87.8% respectively). *C. maximiliani*, *P. kappleri* and *P. macrotis*, on the other hand, have high probabilities of negative effects of moonlight on bat activity (80.4%, 93.8% and 89.5% respectively; Fig. 4).

## DISCUSSION

Here we show that Neotropical bats use habitat above the forest canopy and within the forest understory differently throughout the night. We found that bats are overall more active above the canopy, which is consistent with previous work (*Marques, Ramos Pereira & Palmeirim, 2016*) and that overall bat activity decreases above the canopy throughout the night. We found four species here that were also more common in the canopy (*Cormura brevirostris*, *Molossus molossus*, *Peropteryx kappleri* and *Peropteryx macrotis*). Wing aspect ratios (square of the wingspan divided by wing area) are high for
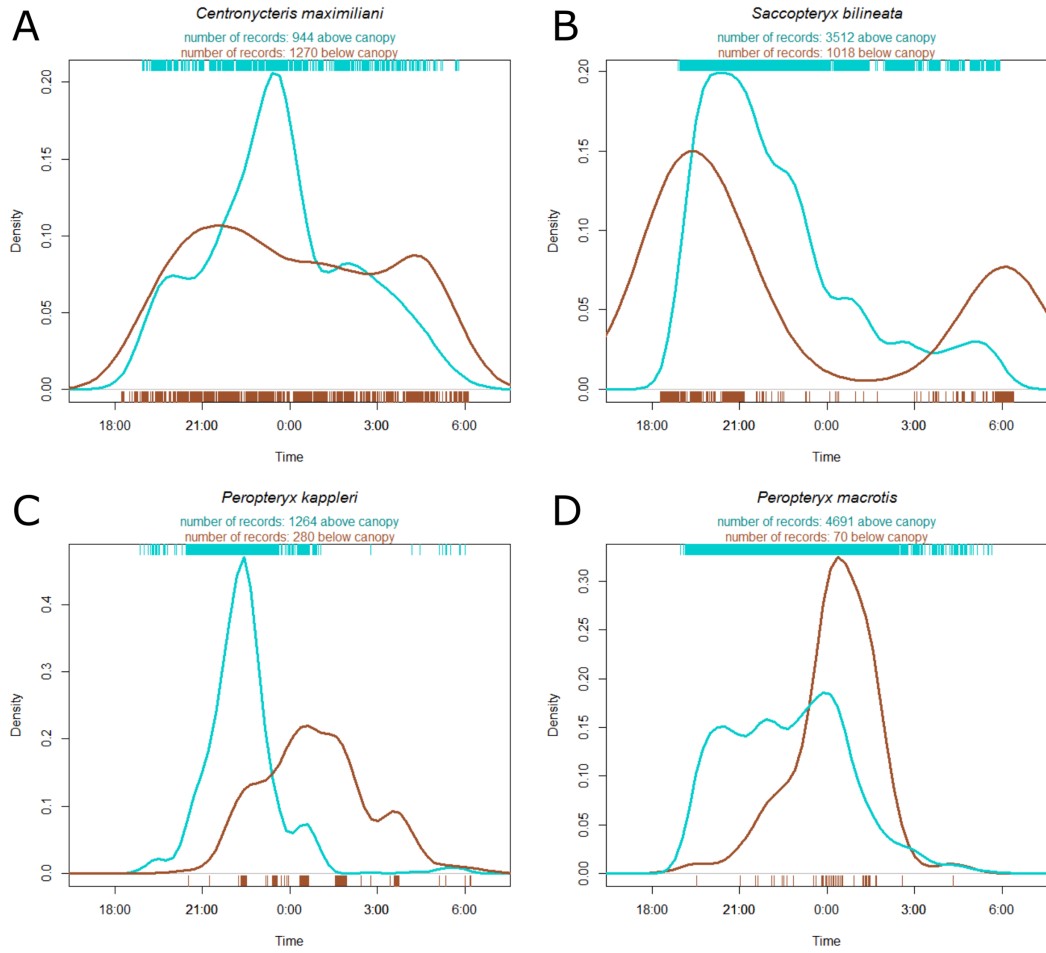

**Figure 3 Kernel density estimate of activity of four most common species recorded at the Nouragues Research Station in French Guiana, plotted by time.** The hash marks at the bottom and top of each plot indicate raw data by understory (brown) and above canopy (blue), respectively. Species names ((A) *Centronycteris maximiliani*; (B) *Saccopteryx bilineata*; (C) *Peropteryx kappleri*; (D) *Peropteryx macrotis*) and individual record numbers for both vertical strata are indicated above the plots. See Supplement for further species plots.

three of these species (*M. molossus*, *P. kappleri* and *P. macrotis*; *C. brevirostris* is not represented in the literature; *Marinello & Bernard, 2014*), suggesting these bats are fast fliers with low maneuverability, which is thought to be advantageous in open spaces, such as above the canopy (see Supplement for exploratory visualization of model estimates by wing aspect ratios and wing loading). Indeed, *P. kappleri* and *P. macrotis* are known edge/open space foragers (*Kalko et al., 2008*; *Barboza-Marquez et al., 2014*). However, we found multiple species that are more active in the understory (compared to one species in *Marques, Ramos Pereira & Palmeirim (2016)*) including strong preferences for understory habitat for *Pteronotus alitonus* and weaker preferences in the same direction for *Pteronotus rubiginosus*. We also found two somewhat conflicting patterns, which might be explained by a lack of species resolution. *Centronycteris maximiliani* in our study weakly preferred the understory, while members of the same genus (*i.e.*, *Centronycteris* sp.) were more common in the canopy in *Marques, Ramos Pereira & Palmeirim (2016)*.

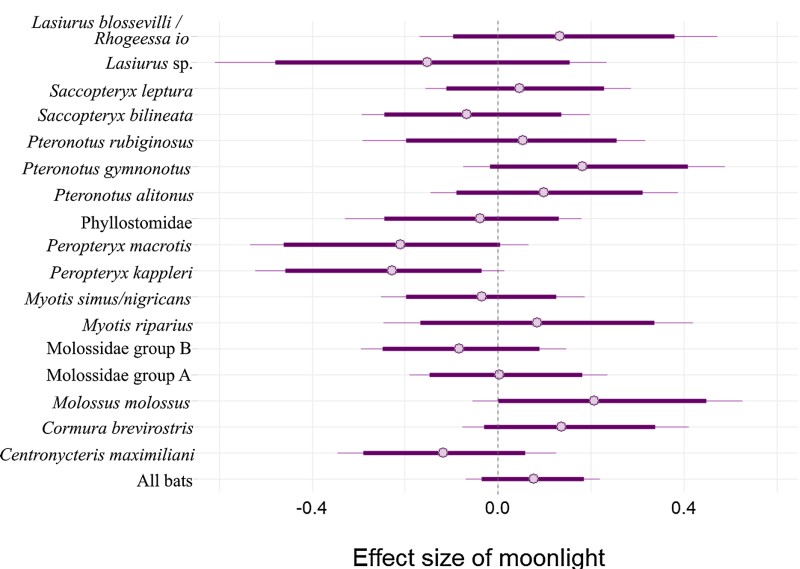

**Figure 4 Model coefficient estimates for activity relative to moon illuminance.** Estimates on left suggest less activity with increasing moonlight, whereas those on right suggest more activity for increasing moonlight. Bold lines are 80% credible intervals, whereas thin lines are 90% CI. See Table 1 caption for acoustic complex species breakdown.

However it is not clear what species might have been included in *Centronycteris* sp. in their study. Similarly, we found a weak preference for the understory in a myotid acoustic complex (*Myotis simus/nigricans*), while *Marques, Ramos Pereira & Palmeirim (2016)* found a canopy preference for *M. nigricans*, although we caution that this previous pattern is informed by only thirteen observations. Both the current study and *Marques, Ramos Pereira & Palmeirim (2016)*, show a clear understory preference for *Myotis riparius*. Other myotid species are thought to prefer to forage in the understory elsewhere in the world (*Kennedy, Sillett & Szewczak, 2014*; *Wellig et al., 2018*), suggesting that this characteristic may be a trait of the genus independent of the geographic location.

For many bats, there were no clear differences in activity between above-canopy and understory habitat (e.g., *Saccopteryx bilineata, S. leptura, Lasiurus* sp. and *Pteronotus gymnonotus*). These patterns may occur for multiple reasons. *Lasiurus* sp., for example, might include multiple species (see Table 1 caption). If some species are more common above the canopy, and others below the canopy, these patterns might be cancelled when analyzed together as an acoustic complex. These patterns instead might occur because bats are just as active in the both vertical strata. *Bernard (2001)*, for example, found the same lack of vertical stratification pattern as we did for *S. bilineata* and *S. leptura*, and the author suggests that this may be because these species fly in large spiral movements occupying both the higher and lower strata. Instead, we found that these two species were more active in the understory early and late in the night, while they were more active above the canopy in the early-middle of the night (Fig. 3B and Supplement). This suggests that these bats roost somewhere near our detectors, likely inside tree cavities and on exposed trunks (*Voss et al., 2016*), but spend the middle hours of the night foraging above the canopy. *S. bilineata* has relatively high wing aspect ratios

(*Marinello & Bernard, 2014*), which is thought to be advantageous for fast flight and confer low maneuverability, yet they spend considerable time below the canopy. This might be because they are opportunistic foragers (*Jung & Kalko, 2011*) that are foraging for different types of insects at different times (*Rydell, Entwistle & Racey, 1996*). However, this is speculation, and a deeper understanding of the natural histories of many of these taxa, along with more morphological data, are necessary for us to pin down exactly what these patterns mean.

Although previous work indicates that bat activity tends to decline with increasing moonlight illumination (*Prugh & Golden, 2014*), here we find a high probability that our sample of Neotropical bats generally show the opposite pattern, increasing bat activity with increasing moon illumination. At the individual species level, *M. molossus, P. gymnonotus* and *C. brevirostris* all show increasing trends with higher levels of moonlight illumination, and all three of these species are more common above the canopy where they are more likely to be exposed directly to moonlight. It is not clear why these bats would prefer moonlight, but it is possible that certain prey are more likely to fly above the canopy on brighter nights (*Roeleke et al., 2018*; *Kolkert et al., 2020*) or that bats are more able to detect predators with vision in moonlight. *C. brevirostris* did not significantly alter activity in moonlight in previous studies (*Appel et al., 2017*), although in one study they trended in the same direction (positively) as found here (*Appel et al., 2019*). *C. maximiliani*, on the other hand, decreased activity in increased moonlight in our study, and is more common below the canopy, where moonlight often fails to penetrate. *P. kappleri and P. macrotis* also both show decreasing activity trends with increasing moonlight, yet they are both more active above the canopy, where moonlight likely plays a larger sensory role. Many species have estimates that substantially overlap no effect. Notable examples are *P. rubiginosus, S. leptura* and *M. riparius*, which all changed activity in relation to moonlight in previous studies (*Appel et al., 2017*, *2019*). *M. riparius*, is a slow-flying bat with a low wing aspect ratio, that likely makes it vulnerable to predation in open spaces, an interpretation shared by authors of previous work that found this bat to avoid moonlight (*Appel et al., 2017*; *Vásquez, Grez & Pedro, 2020*). Thus, it is odd that this species is not affected by moonlight here. *P. rubiginosus* and *S. leptura* both increased activity in moonlight in previous work (*Appel et al., 2019*), but also show no changes here. All three of these species prefer the understory (more strongly in *M. riparius* and more weakly in *S. leptura*), which might suggest that the forest is quite dense at our sites, filtering out most of the moonlight. Such an effect has been shown with respect to artificial light from street lamps (*Straka et al., 2019*). However, as mentioned above *C. maximiliani* was less active in bright nights and also preferred the understory, so the idea that moonlight is filtered out by the canopy is certainly not conclusive.

This study was conducted during the wet season in French Guiana and *Marques, Ramos Pereira & Palmeirim (2016)* occurred during the dry season in Brazil; both studies were short duration (9 and 20 days respectively) and unlikely to offer substantial inference for understanding seasonal effects. Further, many other differences between the French Guiana and Brazilian forests likely obfuscate any speculation about seasonality. Future

research should push to understand vertical stratification over much longer periods of time to understand the effects of seasonality. In addition, a focus on bat prey will likely aid in understanding these patterns. Arthropod prey vary seasonally in their abundance (*Wolda, 1988*; *Lister & Aguayo, 1992*; *Pinheiro et al., 2002*) and those prey likely spend time in different vertical strata (*Schulze, Linsenmair & Fiedler, 2001*). Seasonal changes in arthropod abundances in the Neotropics have been linked to changes in diets of many taxa, including bats (*Lister & Aguayo, 1992*; *Jahn et al., 2010*; *Salinas-Ramos et al., 2015*). Thus, seasonal cycles likely have important consequences for patterns of vertical stratification.

With the constant increase of deforestation of Amazonian primary forests (*Fearnside, 2005*; *Lovejoy & Nobre, 2018*) and consequent loss of vertical stratification of these forests (*Silva et al., 2020*), aerial insectivorous bat activity is likely being affected by forest removal and degradation. Delineating specifically how vertical structure shapes bat communities and activity adds critical insight for ecologists and managers. Here we show that monitoring for bats in one vertical stratum only, or during just the early "golden" hours of the night clearly misses important information.

## CONCLUSIONS

We used passive acoustic monitoring to explore how Neotropical bats use space over time. While bats generally were more active above the forest canopy, we show that individual groups of bats use space differently over the course of a night, and some prefer the understory. Given that most bats were more commonly detected above the canopy, it is possible that we might form erroneous conclusions about the quality of that habitat, or make poor management decisions, if we fail to survey habitat in three dimensions, and for the entire duration of a night. We hope that future work continues to explore how animals and their prey use space throughout the night, and over the course of different seasons, which will surely expand our knowledge of these understudied creatures.

## ACKNOWLEDGEMENTS

We would like to thank the Nouragues research station in French Guiana for access to their facilities and canopy tower system, Cory A. Toth for help deploying bat detectors, J. Conrad, T. Shatwell, and C. Kyba for help quantifying moon illuminance, Diogo Provete, Brock Fenton, Adriana Carolina Acero Murcia, and an anonymous reviewer for substantially improving an earlier version of this manuscript.

### Funding

This work was funded by the CRNS (a 2017 Nouragues Travel Grant to Jesse R Barber). Additional funding provided by NSF (GRFP 2018268606 to Dylan GE Gomes and IOS 1920936 to Jesse R Barber). Giulliana Appel was supported by a Coordenação de Aperfeiçoamento Pessoal Nivel Superior (CAPES) scholarships (Finance code 1) and Sandwich fellowship CAPES Process (88881.362190/2019-0). The funders had no role in

study design, data collection and analysis, decision to publish, or preparation of the manuscript.

## Grant Disclosures
The following grant information was disclosed by the authors:
CRNS.
NSF: GRFP 2018268606 and IOS 1920936.
Coordenação de Aperfeiçoamento Pessoal Nivel Superior (CAPES): Finance code 1.
CAPES: 88881.362190/2019-0.

## Competing Interests
The authors declare that they have no competing interests.

## Author Contributions
- Dylan G.E. Gomes conceived and designed the experiments, performed the experiments, analyzed the data, prepared figures and/or tables, authored or reviewed drafts of the paper, and approved the final draft.
- Giulliana Appel analyzed the data, authored or reviewed drafts of the paper, and approved the final draft.
- Jesse R. Barber conceived and designed the experiments, performed the experiments, authored or reviewed drafts of the paper, and approved the final draft.

## Data Availability
The data and code are available at Zenodo:

Dylan Gomes, & Jesse Barber. (2020). Data and code for: Time of night and moonlight structure vertical space use by insectivorous bats in a Neotropical rainforest: an acoustic monitoring study [Data set]. Zenodo. DOI 10.5281/zenodo.4265167

## Supplemental Information
Supplemental information for this article can be found online at http://dx.doi.org/10.7717/peerj.10591#supplemental-information.

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
