# Peer review of "Time of night and moonlight structure vertical space use by insectivorous bats in a Neotropical rainforest: an acoustic monitoring study"

_PeerJ, doi:10.7717/peerj.10591_

## Round 0.1 · original submission · Major Revisions

I have now received three reviews about your manuscript, all of them specialist in Tropical bat ecology. While all of them were very positive about the results presented, they also pointed out key aspects of the paper that necessarily need to be re-worked before a final decision is made. Each reviewer focused on different aspects of the manuscript, so their comments were really complementary.

Specifically, Adriana Acero pointed out that the description of your data analysis procedures needs to be much more detailed. I agree with her and I also provide a number of suggestions on the PDF attached. Both R2 and R3 agree that the discussion needs major rewriting. Please, pay special attention to their suggestion in that regard.

I also highly suggest that you present your data on hourly activity of the four most common bat species as a kernel density, instead of the methods you used.

I also recommend you to follow the structured Abstract format of PeerJ as you revise your paper. It makes it easy for readers to quickly find the information they need.

·

Basic reporting

This timely article is well written but a few changes might increase its impact.
First, the authors should clearly articulate the hypothesis that guided the work. They then should identify specific predictions they will be able to test with their data. These minor changes will enhance the readability and impact of their research.

The title and the abstract give no indication that the data base was obtained by acoustic monitoring. This is an important omission.

Other than these two points, I enjoyed the manuscript. The results are important and interesting. The data are well presented and analyzed.
thanks for asking me to review it.
Brock Fenton

Experimental design

see above

Validity of the findings

see above

Reviewer 2 ·

Basic reporting

Please, see general comments for the authors.

Experimental design

Please, see general comments for the authors.

Validity of the findings

Please, see general comments for the authors.

Additional comments

The manuscript “Temporal variation in vertical stratification of neotropical bats” (#52673) evaluates bat species activity throughout the night in forest understory compared to that above/in the canopy. This is an interesting, little addressed issue, and the results bring significant novelties on the use of vertical stratum by neotropical rainforest bats. Data collection was based on passive acoustic records during nine consecutive nights using two forest towers for accessing the tall stratum, presumably in pristine Amazon forest, in French Guiana. The experimental design and analyses are well done, with a robust dataset and solid results. However, the introduction and discussion need major reviews for increasing focus and the insertion of the results into the literature, better exploring the findings within the context of the study, as I comment below along with minor appointments.

A first aspect is the lack of presentation of the forest type and description of the study site at all. Although presumably the study was in the Amazon rainforest, only in L183 (discussion) does "Amazon rainforest" appear, but even at this point it is not referring to the location of the current study. I think the title could anticipate the forest type (e.g., “Temporal use of vertical strata by neotropical rainforest bats”), and the methods must begin with the study site description: the general climate, geographical coordinates, the type of forest, conservation status, the height of the canopy and the characteristics of the understory (nearby the towers).

The introduction focuses conservation aspects (first paragraph), the appropriateness of bats as a study model, and the acoustic methods (second paragraph), but lacks expectations about the activity and use of the vertical strata by the bat groups recorded here. It can be substantially improved with the description about what we known on bat species vertical stratification, the activity and use of the strata by different bats, and accounting for the study forest types (it is not currently mentioned even for cited studies). Most of the current content could be reduced or excluded.

The objective, as described in both the abstract (L15-16) and the main text (L51-53), does not clearly present the study question, but it rather brings a description of methods, such as the use of canopy towers, what was monitored and where (above and below canopy), and how long and when (throughout nine nights in the wet season). Thus, I suggest rewriting the objective to clearly show the addressed question, what the authors asked, before presenting how, where or when it was approached.

L16-17: Use the past tense when presenting results. Change “We show that…” to “We found that…”, and the following sentence (L18) to “Overall, bats were more active above canopy, but multiple species/acoustic complexes were more active in the understory”. In this sentence, it is unclear whether “multiple species” means the same as “acoustic complexes”, or it refers to individual species plus those cases recognized as acoustic complexes. Please clarify.

In addition, the tall class of the vertical stratum is unclear as sometimes it is used “above the canopy” and in other sentences is “in the canopy”. Were the recorders installed inside the canopy (just below the taller canopies) or indeed above canopy (which means the open space above the forest)? This is important for understanding the results, because the exposure of bats in the environment (e.g., moon light), the types of resources used, and the structural/physical complexity of the environment influences the bat activity. I think that this use of ‘above’ or ‘in’ the canopy should be consistent throughout the manuscript, and precisely described in methods.

L19: Change to past tense.

L22: Data collection is restricted to nine consecutive nights only, so it seems inappropriate to address variation between seasons as part of the conclusion. The focus should be on the variation over night between the two strata, avoiding speculations out of this context, like seasonal differences, both in the abstract and the discussion.

L58: include the distance between the towers.

L63: Four recorders were used in total, correct? What is the approximate maximal distance that recorders can detect bat calls? Is it similar above and below the canopy?

L76-77: Phyllostomidae is also in Table 1.

L116: the first and second sentences are repetitive; they could be just one, e.g., “Overall bat activity was estimated to be 9.5 times higher in the canopy than in the understory.”

L119-120: change “are/aren’t” to “were/were not”

L123: This result (a positive effect of moonlight on overall bat activity) was not discussed; how moonlight affects activity below and above canopy would be additionally applicable and interesting into the focus of the manuscript.

L136-138: change “is” to “was”, “they are” to “it was”, and “has” to “had”

L140-141: change “are” to “were”, and “is” to “was”

L145-150: The beginning of the discussion looks like an introduction. I suggest starting with the main/most general findings supported by the results, and then use the literature to introduce the main general discussion.

L151: This paragraph starts saying that the greater activity in the canopy corroborates the work of Marques et al. (2016), but the following paragraph (L158) starts pointing out "some differences" with the study by Marques et al. (2016). It is unclear what these differences are, and it appears contradictory to the previous text.

L158-159: Even though Marques et al. had sampled in a different season, the current data from nine nights are no longer enough to represent one season, which makes all the following discussion difficult.

L169: At this point in the discussion, it should be fine to address which groups of species have similar and different patterns, which are possible differences that may explain the patterns (e.g., feeding habit, foraging behavior). “No clear preference for stratum” is also a pattern. It sounds to me a very interesting result that deserves a more appropriated and detailed discussion, using a richer literature.

L174-182: This part is a bit confused as only Saccopteryx spp. are used as examples for approaching a general pattern; and partition of nighttime or canopy strata is also unclear without a previous discussion on time and space use by the bat groups.

L191-194: estimating distance to the roosts is not central in the paper. Even speculative parts should be closer to the major paper’s aims.

L199-201: This sentence is too strong. Even considering variations in activity over night and strata, conclusions about habitat quality would not necessarily be erroneous. In fact, contrary to this idea, if most species use both strata, then sampling only the understory seems enough for detecting most of the bat fauna.

Figure 1: I suggest placing the species (X-axis) in decreasing or increasing order of effect size. This will allow to quickly identify which groups have similar and opposite trends.

Figure 2: I suggest the same, to order the spp. vertically according to the effect sizes, taking the values for understory (though the values for canopy will not be ordered, it will improve visualization of differences between spp. and strata).

Table 2: Do no italicize the family names, and “sp.”

·

Basic reporting

The purpose of the manuscript “Temporal variation in vertical stratification of neotropical bats” was to study the vertical space used by neotropical bats in French Guiana. The authors used GLMMs with a Bayesian approach and passive acoustic tools to achieve this goal. The calls were recorded in two localities (towers), each one sampling the two strata. The authors found that the bats were more active in the canopy than understory. With intervals of 80%, it was evidenced that six species are more active in the canopy, five in the understory, and other six without preference strata. They mention that lunar light had a positive effect on activity but that the data cannot be explored for the species. Overall bat activity decreased every hour in the canopy as the night progressed, whereas activity in the understory did not change over time. This study aims to contribute to the knowledge about the temporal and special segregation of bat communities in neotropical forests.

The literature is adequate, but I suggest strengthening the introduction and discussion, as well as including the hypotheses and predictions in the introduction, and how these will be tested.

The supplementary material data must have a more detailed description because they must be self-explanatory for whoever wants to use this information in the future. For example, in xls data, the variables are abbreviated, and it doesn't have units. What does mean x, color, number, S. Alt, M.Az, T.illm, what are the units of these variables?
Minor comments and others are in the attached file.

Experimental design

Given that the authors want to work a new statistical approach to what is commonly worked, I suggest that the authors writing why they choose this new approach and why this approach would allow us to understand the segregation processes in bats. I suggest that the methods should be rewritten, and data treatment in detail (view: Zuur and Ieno. 2016; Stanton 2017). For example, describe whether the data was transformed, how many models are tested, and what was the model selection criterion. Include the statistical output of the models is reported, or of the fixed effect (lunar luminosity).

Also, the authors need to include how the moonlight measurement was obtained in the study area. The cited article does not refer to a specific effective measurement and if to the need to have several measurement indices of this variable. So I suggest that the authors be more specific and mention how they obtained this variable. Even in the data set (supplementary material) a column with the name color (red, green) that makes me doubt if this is the variable that was used as luminosity.

Another important aspect to be included is the characterization of the habitat, since the forest cover around the sensors can influence in the recording of the species (Gibb et al. 2018). More information about the habitat would resolve this question.

In lines 151-152, 153, 155, 171-172 the statements are very general, so they are not informative. I suggest that the authors be more specific, and include the name of the species or acoustic complexes.

The discussion should be strengthened, specifically in temporal and spatial segregation in the study area, and not discuss strongly topics conservation and seasonality aspects, because you did not have data in the study to support this discussion.

Validity of the findings

The authors found differences in the use of the strata for some species. As well as temporal segregation was found for in four species of the Emballonuridae family selected for this analysis. But there are gaps in the methodology and in the statistical analysis that do not allow us to understand why the authors selected 4 species of the 19 sonotypes that they registered. The authors mention that there is a deficit in the sampling (n samples size), but I think that the data could be include, or maybe do a statistical analysis to support to discard the other species.

I suggest, that authors exploring in the discussion the presence of the species in each stratum, this related to their body morphology to explore or find resources in those environments. For example, narrow wings might move more easily on the understory than on the canopy. Another important aspect is that the authors do not mention how they obtain the luminosity measurement and this variable is included in the models, and have positive results in the activity.

Additional comments

The manuscript is interesting and highlights the importance of this study on spatial and temporal segregation in Neotropical bats. I suggest improving the methodology and statistical design. Also, is very important to include the hypothesis and the prediction that the authors want to test. As well as take advantage of the suggested topics in the introduction and the discussion section (partition of the environment, the temporality of the resources (insects) at night, the luminosity and its influence on the activity in the bats).

---

## Round 0.2 · accepted · Accept

It's my pleasure to accept this version to publication. The two reviewers made their final suggestions and corretions that could be added to the manuscript in the proof review stage.

Reviewer 2 ·

Basic reporting

See comments for the authors.

Experimental design

See comments for the authors.

Validity of the findings

See comments for the authors.

Additional comments

I really liked the review carried out by the authors, with energy to approach very well the points raised on the first version, incorporating modifications that bring up the main findings in more depth, and the use of appropriate literature. It was a pleasure to participate in the review, and I believe that this paper will have a great audience. My only suggestion about the content is to change the current sentence at the end of the Abstract (Discussion), as the main message is not exactly what is there, but rather the conclusions available in the two first sentences of the Discussion (and in the Conclusion) of the main text.

In addition, some typos should be corrected during the editorial process, for instance: period (L 112), commas (L 231, 243), italic/non-italic for “i.e.” and “sp.” (L 205, 242, 244), numbers up to 9 in full (L 147), and "Pteronotus" (L 240). I also suggest avoiding abbreviation of species names (genus) at the beginning of phrases (L 211, 214, 265, 280, 283). Increasing font size within figures 2 and 3 will improve visualization, mainly the titles and axes’ names for Kernel scatterplots.

·

Basic reporting

The manuscript "Time of night and moonlight structure vertical space use by insectivorous bats in a Neotropical rainforest: an acoustic monitoring study", is clear, well structured, and consistent with the scope of PeerJ.

This study aims to contribute to the knowledge about the temporal and special segregation of bat communities in neotropical forests of French Guiana.

The figures and tables are clear and appropriate.

The authors applied the suggestions of the reviewers, the adjustment of the text was assertive, allowing for a more pleasant reading without information gaps. The methods allow the replicability of the study. The cites and references are complete and adequate.

Experimental design

The methods improved a lot.

The data collection and the analysis description is good.

The statistical output of the models was reported, but I suggest it like supplemental material.

Also, I suggest adding the ethical note that is missing and move the moonlight data for the methods section (See minor comments).

Validity of the findings

This study shows pertinent results about the spatial and temporal distribution of bats in the tropical forest through passive acoustic monitoring.

Additional comments

The new manuscript is really good, and clear. The methods section improved a lot and It is now clearer for the reader.

The data collection and the description of statistical analysis are well detailed. The discussion is adequate and in accordance with the results found.

I have some minor considerations like moving the “Moonlight data collection” to the methods section, add the ethical note, and the data availability DOI or link to Zenodo (See PDF file).

Congratulations by the new version.